# CRISPR DNA Base Editing Strategies for Treating Retinitis Pigmentosa Caused by Mutations in *Rhodopsin*

**DOI:** 10.3390/genes13081327

**Published:** 2022-07-26

**Authors:** Maria Kaukonen, Michelle E. McClements, Robert E. MacLaren

**Affiliations:** 1Nuffield Department of Clinical Neuroscience, University of Oxford, Oxford OX3 9DU, UK; maria.kaukonen@ndcn.ox.ac.uk (M.K.); michelle.mcclements@eye.ox.ac.uk (M.E.M.); 2Oxford Eye Hospital, Oxford University Hospitals NHS Foundation Trust, Oxford OX3 9DU, UK

**Keywords:** inherited retinal dystrophy, retinitis pigmentosa, rhodopsin, *RHO*, base editing

## Abstract

Retinitis pigmentosa (RP) is the most common group of inherited retinal degenerations and pathogenic variants in the *Rhodopsin* (*RHO*) gene are major cause for autosomal dominant RP (adRP). Despite extensive attempts to treat *RHO*-associated adRP, standardized curative treatment is still lacking. Recently developed base editors offer an exciting opportunity to correct pathogenic single nucleotide variants and are currently able to correct all transition variants and some transversion variants. In this study, we analyzed previously reported pathogenic *RHO* variants (*n* = 247) for suitable PAM sites for currently available base editors utilizing the *Streptococcus pyogenes* Cas9 (SpCas9), *Staphylococcus aureus* Cas9 (SaCas9) or the KKH variant of SaCas9 (KKH-SaCas9) to assess DNA base editing as a treatment option for *RHO*-associated adRP. As a result, 55% of all the analyzed variants could, in theory, be corrected with base editors, however, PAM sites were available for only 32% of them and unwanted bystander edits were predicted for the majority of the designed guide RNAs. As a conclusion, base editing offers exciting possibilities to treat *RHO*-associated adRP in the future, but further research is needed to develop base editing constructs that will provide available PAM sites for more variants and that will not introduce potentially harmful bystander edits.

## 1. Introduction

Retinitis pigmentosa (RP) is the most common group of inherited retinal degenerations affecting approximately 1 in 4000 people worldwide [1]. According to the Retinal Information Network (http://sph.uth.edu/retnet, accessed on 28 June 2022), mutations in 23 genes have been linked to autosomal dominant (ad) RP. Mutations in the *Rhodopsin* (*RHO*) gene are a major cause of adRP with over 150 pathogenic variants reported that collectively account for 30–40% of all adRP cases [2,3].

The *RHO* gene encodes the rod photoreceptor-specific photopigment that is responsible for scotopic photoreception. While a great majority of reported *RHO* variants are associated with adRP, some are also reported to cause autosomal dominant congenital stationary night blindness (adCSNB) [4,5] and two mis-sense variants have been associated with recessive RP [6,7]. *RHO* variants can be classified, based on their exact pathologic mechanism, into at least seven discrete classes [8], but typically recessive variants result from loss of function and dominant variants from gain of function and/or dominant negative activity. Pathogenic variants in *RHO* cause a classical form of RP, in which rod photoreceptor dysfunction and degeneration cause night blindness and loss of peripheral vision, typically within the first two decades of life, whereas central vision is preserved until the end stages of the disease [1]. Spectrum-frequency and genotype-phenotype correlation of *RHO* variants have recently been reviewed by Luo et al. [9].

Despite extensive attempts to treat *RHO*-associated adRP [10,11,12,13], standardized curative treatment is still lacking. Gene replacement therapy has shown great results in treating recessive *RPE65*-associated Leber congenital amaurosis type 2 [14], however applying such an approach to dominant-negative diseases is limited by the requirement for silencing the defective allele.

Recently developed CRISPR DNA base editors offer an exciting opportunity to correct pathogenic single nucleotide variants (SNVs) in dominant diseases. In principle, base editors consist of a deactivated Cas protein that is linked to a deaminase enzyme, forming together a construct that can introduce irreversible A>G (adenine base editors, ABEs) [15], C>T (cytosine base editors, CBEs) [16] or C>G (glycosylase base editors, GBEs) [17] edits. In contrast to traditional CRISPR/Cas systems, the endonuclease domain of Cas proteins in base editing constructs have been deactivated/changed to nickase, and thus base editors do not introduce double-stranded DNA breaks, making base editing a safer approach for gene editing in vivo [16]. Additionally, conventional genome editing utilizing active Cas endonucleases rely on homology directed repair (HDR) that is not ideal for non-dividing cells such as photoreceptors [18], whereas successful base editing does not require cell division. Despite lost endonuclease activity, the deactivated Cas protein in base editors is still able to bind DNA, in the presence of a construct-specific protospacer adjacent motif (PAM) and is targeted to the desired genomic position with a guide RNA molecule that binds to the complementary strand of the target nucleotide [16]. As DNA base editors can edit both forward and reverse strands, all pathogenic transition variants (A>G, G>A, C>T, T>C) and even some transversion variants (C>G, G>C) can in theory be corrected with them, collectively enabling correction of more than 61% of all reported human pathogenic variants [19]. Different base editing constructs and their preferences have been reviewed recently by Kantor et al. [20].

In this article, we analyzed all disease-associated *RHO* variants for their editability with the most commonly used ABEs, CBEs and GBEs, and designed suitable guide RNA sequences for each editable variant, thus forming the basis for future in vitro and in vivo experiments aiming to develop treatment strategy for *RHO*-associated adRP with base editing. 

## 2. Materials and Methods

All *RHO* variants in the publicly available Leiden Open Variation Database (LOVD) 3.0 (http://www.lovd.nl, accessed on 19 April 2022) [21], ClinVar (http://www.ncbi.nlm.nih.gov/clinvar, accessed on 19 April 2022) [22] and Genome Aggregation Database (gnomAD) v2.1.1 (http://gnomad.broadinstitute.org, accessed on 19 April 2022) [23] were downloaded on 19 April 2022 using the GRCh37/h19 genome build as the reference genome and ENST00000296271 as the reference transcript. First, each dataset was analyzed separately to characterize and compare their contents. The variants were divided based on the reported:variant type into insertions/deletions/duplications (indels, not editable with base editors) and SNVs (possibly editable with base editors);predicted variant consequence into nonsense, missense, synonymous, splice site (+/− 2 nucleotides from exon-intron boundaries), intronic, 3′ and 5′ untranslated region (UTR), stop loss and start loss variants; and byclinical interpretation of the variant pathogenicity into pathogenic/likely pathogenic, variant of unknown clinical significance (VUS), conflicting interpretations of pathogenicity and benign/likely benign variants.

Lastly, pathogenic/likely pathogenic variants per dataset were analyzed for editability by ABE (G>A, C>T variants), CBE (T>C, A>G) and GBE (G>C, C>G).

In practice, not all of these variants can be corrected with currently available base editors, as successful base editing also requires a base editor-specific PAM site to be available adjacent to the targeted nucleotide in a position where it will place the variant within the construct’s editing window. To assess actual editability of the disease-associated *RHO* variants, datasets from all the three databases were merged, and duplicate entries removed. Variants labelled in at least one of the databases as pathogenic or likely pathogenic were included in the subsequent assessment of amenability to base editing. Variants were then grouped into indels and SNVs, and SNVs editable with ABEs (G>A, C>T), CBEs (T>C, A>G) or GBEs (C>G, G>C) were labelled for subsequent PAM site screening.

For the PAM site analysis and guide RNA design, altogether four ABEs (Figure 1a) (SpCas9-ABE8e [24], SaCas9-ABE8e [24], KKH-SaCas9-ABE8e [24], CasMINI-ABE8e [25]), two CBEs (Figure 1b) (SpCas9-CBE (BE4) [26], SaCas9-CBE (SaBE4) [26]) and one GBE (Figure 1c) (SpCas9-GBE (GBE2.0) [27]) were selected, representing the most commonly used base editors. PAM site screening and guide RNA design were completed using the Benchling software (San Francisco, CA, USA). As PAM site requirements, editing windows and optimal guide lengths vary between different base editors (Figure 1a–c), including all the above-mentioned constructs in the analysis increases possibilities to identify a suitable construct for each targeted *RHO* variant. As an example, base editors utilizing *S. pyogenes* Cas9 (SpCas9) require an ‘NGG’ PAM site downstream the target site and edits bases 4-8 [24,26], whereas *S. aureus* Cas9 (SaCas9) recognizes ‘NNGRRT’ (R=G/A) sequence [24,26] and the *S. aureus* KKH strain (KKH-SaCas9) more relaxed ‘NNNRRT’ PAM sequence [24], while exhibiting efficient editing in wider editing window (Figure 1). Currently, GBE is only available with SpCas9 [27], and thus editable transversion variants (G>C, C>G) were only screened for the ‘NGG’ PAM sites.

Finally, each designed guide was analyzed manually for likely bystander edits, which are unwanted edits that may occur if the editing window includes one or more additional identical bases to the aimed target base.

## 3. Results

### 3.1. LOVD

The LOVD dataset included altogether 257 unique *RHO* variants reported from 906 individuals (Appendix A). The majority (92%) of the reported variants were SNVs, while 7.8% were indels (Figure 2a). Missense variants accounted collectively for 80% of all the SNVs. Clinical interpretation was reported to all except one variant, with 67% of all the unique variants interpreted as pathogenic or likely pathogenic variants (Figure 2b). Of these variants, altogether 59% could in theory be edited with ABE, CBE or GBE (Figure 2c).

The most common variant in this dataset was the c.1040C>T, p.Pro347Leu with 95 entries (Figure 2d). This variant is interpreted as pathogenic or likely pathogenic and could be edited, on the reverse strand, with ABE. Collectively, the ten most frequently reported variants covered 39% of all the 906 entries and all of them had been labeled pathogenic or likely pathogenic at least once, although four had reportedly conflicting interpretations. Nine of these variants are editable transition or transversion variants, with the c.68C>A, p.Pro23His being the only one that cannot be corrected with current base editors (Figure 2d).

### 3.2. ClinVar

The ClinVar dataset included altogether 452 unique *RHO* variants (Appendix A). Again, the majority (93%) of the reported variants were SNVs, and amongst them, the most common variant type was missense variant (61%) (Figure 3a). Clinical interpretation was reported to all except one variant, and the two most common interpretations were VUS (39%) and pathogenic or likely pathogenic variant (33%) (Figure 3b). Of the latter, altogether 57% could in theory be edited with ABE, CBE or GBE (Figure 3c), with variants editable with ABE forming again the largest subgroup amongst the editable variants. Associated phenotype information was provided for 104 of the 148 pathogenic or likely pathogenic variants, with the most common reported phenotype RP4 (*n* = 73), RP (*n* = 12) and retinal dystrophy (*n* = 10), in addition to seven notations of adCSNB and single reports of congenital hypomyelinating neuropathy 2 and retinitis punctata albescens.

### 3.3. gnomAD

The gnomAD dataset included altogether 561 *RHO* variants (Appendix A), of which 546 were SNVs and 15 indels (Figure 4a). Amongst the SNVs, missense variants were again the most common variant type with 40% share. Clinical interpretations were provided for only 25% of all the variants, with 3.4% labeled as pathogenic or likely pathogenic (Figure 4b). Of these, 79% were editable transition or transversion variants (Figure 4c). Of the pathogenic or likely pathogenic variants, the c.448G>A, p.Glu150Lys had the highest allele frequency (0.000048) and thus could be regarded as the most common disease-associated *RHO* variant in the gnomAD dataset. Interestingly, this variant was only identified in the South Asian sample set (allele count 12/30,616), while it was absent in the other cohorts. As a G-to-A transition, this variant could be edited with ABE. 

### 3.4. PAM Site Screening and Guide RNA Design

Based on the LOVD, ClinVar and gnomAD datasets, the majority of pathogenic or likely pathogenic *RHO* variants could be edited with ABE, CBE or GBE as judged by variant types. However, successful base editing also requires an available PAM site that places the target nucleotide into the construct’s editing window and ideally no unwanted bystander edits would be introduced. To assess the practical possibilities to edit disease associated *RHO* variants, we combined data from all the three datasets and included in the subsequent analysis all unique variants that were labeled as pathogenic or likely pathogenic in at least one of the datasets. As a result, 247 unique variants were listed, of which 36 were indels, and thus not editable with BE, and 211 SNVs. Of the SNVs, 75 were not editable transversion variants, while, in theory, 65 were editable with ABE, 41 with GBE and 30 with CBE (Figure 5a).

We then analyzed these 136 variants for available PAM sites and designed guide RNAs using the most commonly utilized base editors including four ABEs, two CBEs and a GBE (Figure 1). As a result, 43 variants could be targeted with these constructs (Appendix A). Of the 65 variants potentially editable with ABEs, 32 could be targeted with at least one of the constructs—and 19 with multiple constructs (Figure 5b, Appendix A). Amongst the ABE-constructs, SpCas9-ABE PAM sites were found most commonly (*n* = 25), whereas the CasMINI PAM site (TTTR) with narrow editing window seemed very restricted and no PAM sites were found for that construct (Appendix A). Of the CBE-targetable variants (*n* = 30), guide RNAs could be designed for eight variants and for GBE-targetable variants (*n* = 41) for only three variants (Figure 5b, Appendix A).

As bystander edits form an important safety concern for base editing, we next analyzed all the designed guide RNAs for likely bystander edits. This revealed that all the SaCas9-ABE8e and -CBE as well as all the SpCas9-CBE guides will likely introduce additional unwanted edits, as will also 74% of the KKH-SaCas9-ABE8e and 60% of the SpCas9-ABE8e guides (Figure 5c, Appendix A). In contrast, none of the SpCas9-GBE guides were predicted to create bystander edits. Not all bystander edits will necessarily be harmful, e.g., if they do not introduce amino acid changes (silent variants) or defects in splicing. Regrettably, the majority of predicted bystander edits would likely introduce amino acid changes (Figure 5c, Appendix A), and in the case of SaCas9-ABE8e, KKH-SaCas9-ABE8e and SaCas9-CBE, the majority of guides with predicted bystander edits would likely change multiple amino acids.

## 4. Discussion

Pathogenic *RHO*-variants explain a large proportion of adRP cases [2,3], but despite extensive attempts, a standardized curative treatment is still lacking. As recently developed CRISPR-Cas DNA base editors can correct all transition [15,16] and even some transversion variants [17], it might be applicable method to treat *RHO*-associated adRP. In this study, we analyzed disease-associated *RHO*-variants for their amenability to DNA base editing. In summary, 55% of the pathogenic or likely pathogenic *RHO*-variants could be corrected with base editors if judged by the variant types, however, PAM site availability and high incidence of likely bystander edits pose important limitations to the method’s current translational potential.

The *RHO* gene was one of the first to be implicated in RP and variants in it are estimated to explain 30–40% of all adRP cases [2,3]. To analyze base editing options for as many disease-associated *RHO*-variants as possible, we downloaded all *RHO*-variants from three publicly available databases, namely the LOVD [21], ClinVar [22] and gnomAD [23]. LOVD and ClinVar are so called locus-specific databases that store information on variants and their observed phenotypic consequences, and hence mainly include data from affected individuals. In contrast, gnomAD aggregates variation data from multiple large sequencing projects, and therefore illustrates normal genetic variation, but also provides less biased information on variant prevalence. As expected, pathogenic or likely pathogenic variants were more commonly reported in the LOVD and ClinVar datasets (representing 67% and 33% of all the reported variants, respectively), whereas in gnomAD, this label was given only to 3.4% of the variants—and no clinical interpretation was provided for 75% of the variants. Many disease-associated variants were reported in only one of the datasets, which is illustrated by the number of unique pathogenic/likely pathogenic variants per each dataset (*n* = 19–172) versus unique variants in all datasets combined (*n* = 247). Each dataset has also slightly different accompanying information for the variants, such as population-specific allele counts in the gnomAD, readily listed original publication references in LOVD and clearly listed phenotypic information in ClinVar. As a conclusion, including variant data from multiple databases is favorable when analyzing editing options for disease-associated variants.

The most common disease-associated *RHO*-variant in the LOVD dataset was the c.1040C>T, p.Pro347Leu, which alone covered over 10% of all entries. This variant has previously been reported to be the most common pathogenic *RHO*-variant in Spain [28] and France [29], and has also been reported in the UK [30] and China [31]. In contrast, the p.Pro347Leu was reported only once in the gnomAD dataset and with an allele frequency (AF) of 0.000032, whereas the most common disease-associated variant was the c.448G>A, p.Glu150Lys (AF = 0.000048). As transition variants, both could be corrected with base editing, as could all but one of the ten most commonly reported disease-associated variants in the LOVD dataset (Figure 2d). Indeed, the only non-editable variant amongst them was the c.68C>A, p.Pro23His that, as a C-to-A transversion, cannot be reverted with current base editors. This variant is thought to be the most common pathogenic *RHO*-variant in North America [32,33], but interestingly is not reported in the gnomAD dataset at all (Appendix A).

Judging solely on the variant type, base editing appears to be a very promising treatment option for *RHO*-associated adRP: the majority of pathogenic or likely pathogenic variants could be edited with either an ABE, CBE or GBE (Figure 5a)—and, as described above, the majority of the most common disease-associated variants are editable. Of the different base editors, the ABE was the most commonly required construct to correct *RHO*-variants—similar to what has previously been observed for pathogenic *CRB1* [34] and *ABCA4* [35] variants. The C-to-T transition variants are particularly common in the human genome, as cytosines and 5-methylcytosines are spontaneously deaminated into uracils and thymines, respectively, with an estimated occurrence of 100–500 times per cell per day [15,36]. The C-to-T transitions are estimated to account for approximately half of all known pathogenic SNVs, and as they can be edited in the reverse strand with an ABE [15], it is not surprising that ABEs are frequently reported to be the most commonly usable construct to correct pathogenic variants.

In addition to suitable variant type, successful base editing also requires a construct-specific PAM site to be available near the target variant. PAM site availability is limited by the length and complexity of the required sequence: e.g., for the ABE-editable *RHO*-variants, an SpCas9 PAM site was found to be available for 38% of the variants, while the more restrictive SaCas9 PAM site was available for only 9.2%. Overall, a PAM site was most likely found to be available for the ABE-editable and least likely for the GBE-editable variants (Figure 5b). These differences are likely explained by the fact that there are more ABE-constructs available, with overlapping PAM site requirements, whereas in the case of GBE, there are currently only constructs utilizing the SpCas9 [17,27]. In future, PAM site availability will likely be less restrictive and more variants will become targetable, as new CRISPR Cas nucleases, each with their own PAM site requirements, are found with an increasing pace. As an example, near-PAMless constructs such as the SpG and SpRY strains of *S. pyogenes*, with more relaxed NGN and NRN/NYN PAM site requirements have been recently developed [37], however delivering these constructs in vivo will face similar packaging-size related challenges than the other SpCas9-based constructs do (discussed in more detail below).

Strict PAM site sequence requirements could be compensated by wider editing windows. On the other hand, a wider editing window increases the risk of unwanted bystander edits. We observed likely bystander edits for the majority of all designed guide RNAs in all constructs except the SpCas9-GBE (Figure 5c), which has a very narrow editing window. Of the ABEs, likely bystander edits were predicted for 60% of the SpCas9 gRNAs, whereas for the SaCas9 and KKH-SaCas9 guides, with wider editing windows, bystander edits were precited for 100% and 74%, respectively. Not all bystander edits are necessarily harmful, as they can also result in silent variants. Unfortunately, only 0–18% of the bystanders, varying by the construct in question, were predicted to be silent variants. Wider editing windows were also associated with a greater number of bystander edits: the majority of SpCas9-ABE bystanders changed one amino acid, whereas in the case of SaCas9-ABE and KKH-SaCas9-ABE, the majority of guide RNAs were predicted to result in bystander edits involving multiple amino acids. Some amino acid changes might be tolerated, but extensive variant-, guide- and construct-specific analyses need to be performed to confirm safety of any given base editing treatment.

Different base editing constructs differ also in the terms of size, which is important when choosing the delivery method to target the photoreceptor cells. To date, recombinant vectors based on adeno-associated virus (AAV) are the preferred delivery method to target photoreceptors, as they have shown excellent safety and efficacy profiles in humans [38]. In regards to base editors, the AAVs’ limited cargo capacity (~5 kb) [39] poses challenges as the SpCas9 base editors do not fit into a single AAV [24,26] and the smaller SaCas9-based ones are at the border of being too large [24,26]. Dual AAV vector strategies to target photoreceptor cells have shown promising results in terms of increasing the accepted cargo size, however the transduction efficiency might be lower [40]. Lentiviruses represent another frequently used viral delivery method and have the advantage of larger cargo capacity (~8 kb) [38]. They have recently been used to deliver successfully an SpCas9-ABE to correct *Rpe65* nonsense variant in vivo [41], however their translational applicability to patients might be limited by their potential insertional mutagenesis [38]—and in terms of targeting the photoreceptor cells, by their reduced transduction efficiency [42]. In future, non-viral delivery methods such as minicircles [43], gold nanoparticles [44] or engineered virus-like particles [45] might become increasingly usable methods. In terms of packaging size, smaller Cas nucleases, such as the recently discovered hypercompact CasPhi with a very flexible PAM site (TBN) [46] or the *Staphylococcus lugdunensis* nuclease (NNGG) [47] might offer important improvements to the field, but thus far corresponding base editors have not been developed to our knowledge.

In addition to DNA base editing, multiple other technologies to target dominant mutations are also being developed. These include for example prime editing, the newest member of the CRISPR-gene editing toolkit that can correct all SNVs and indels, but currently produces significantly less efficient editing compared with DNA base editing and will not fit into a single AAV [48]. Another potential approach to treat adRP resulting from dominant *RHO* mutations is the dual AAV-based “knock-out-and replace” of *RHO* method, in which the endogenous *RHO* is silenced with active SaCas9, while a codon optimized *RHO* transgene is simultaneously delivered [49]. An important advantage in this method is that it is mutation-independent, however, potential challenges include possible safety concerns related to introducing double-stranded DNA breaks as well as the risk of inducing toxic effects by overexpressing the *RHO* transgene [50]—two important challenges that do not concern DNA base editing.

## 5. Conclusions

In conclusion, DNA base editing has important mechanistic and functional advantages and could, *in theory*, be used to correct the majority of disease-associated *RHO*-variants. However, practical implications are currently limited by PAM site availability, required viral packaging size capabilities and high incidence of likely bystander edits that could abrogate the treatment’s benefit. Further development is needed before DNA base editing can be widely used to correct *RHO*-mutations in adRP patients.

## Figures and Tables

**Figure 1 genes-13-01327-f001:**
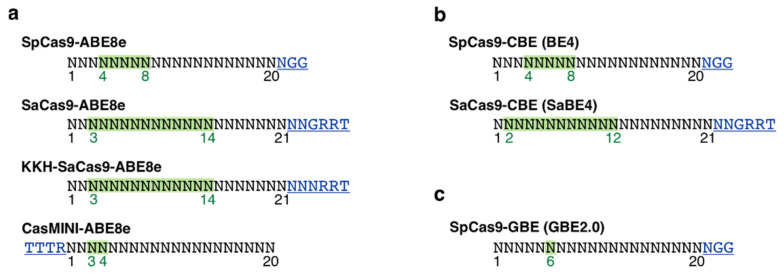
Schematic representation of the (**a**) ABEs, (**b**) CBEs and (**c**) GBEs included in the PAM site analysis and guide RNA design. PAM site requirements (underlined and marked in blue), ideal editing windows (highlighted in green) and optimal guide lengths are construct-specific [24,25,26,27]. R = G/A.

**Figure 2 genes-13-01327-f002:**
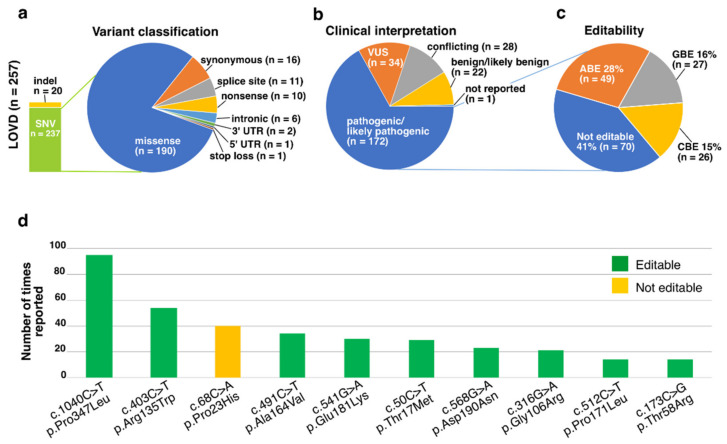
Graphical illustration of the LOVD dataset. (**a**) *RHO* variants from altogether 906 individuals have been reported to the LOVD, comprising 257 unique variants. The most common variant type was missense variant, which covered over 80% of all the reported unique SNVs. (**b**) Clinical interpretation was reported for all but one of the variants, and 67% were labeled as pathogenic or likely pathogenic. (**c**) Of the pathogenic or likely pathogenic variants, 59% could be edited with ABE, CBE or GBE, with ABE being the most frequently usable base editor. (**d**) Ten most frequently reported *RHO* variants in the LOVD dataset were all associated with disease, and together accounted for 39% of all the dataset entries. Only one of these variants, the c.68C>A, p.Pro23His, is not editable with current base editors.

**Figure 3 genes-13-01327-f003:**
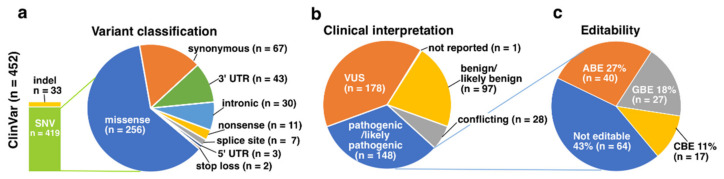
Graphical illustration of the ClinVar dataset. (**a**) Altogether 452 unique *RHO* variants have been reported to ClinVar, with SNVs forming the largest proportion of them with 93% share. Again, the most common variant type was missense variants, which covered over 61% or all the reported unique SNVs. (**b**) Clinical interpretation was reported for all but one of the variants, and 33% were labeled as pathogenic or likely pathogenic. (**c**) Of the pathogenic or likely pathogenic variants, 57% could be edited with ABE, CBE or GBE, with ABE being again most frequently usable option.

**Figure 4 genes-13-01327-f004:**
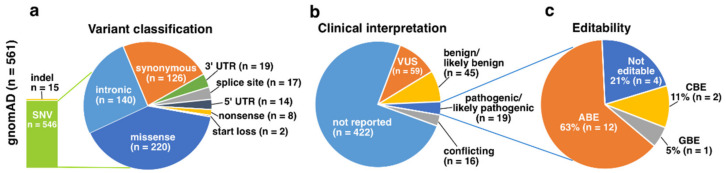
Graphical illustration of the gnomAD dataset. (**a**) Altogether 561 unique *RHO* variants have been reported in gnomAD, with 97% of them SNVs. (**b**) Clinical interpretation was reported for only 25% of the variants, and 3.4% were labeled as pathogenic or likely pathogenic. (**c**) Of the pathogenic or likely pathogenic variants, 79% could be edited with ABE, CBE or GBE.

**Figure 5 genes-13-01327-f005:**
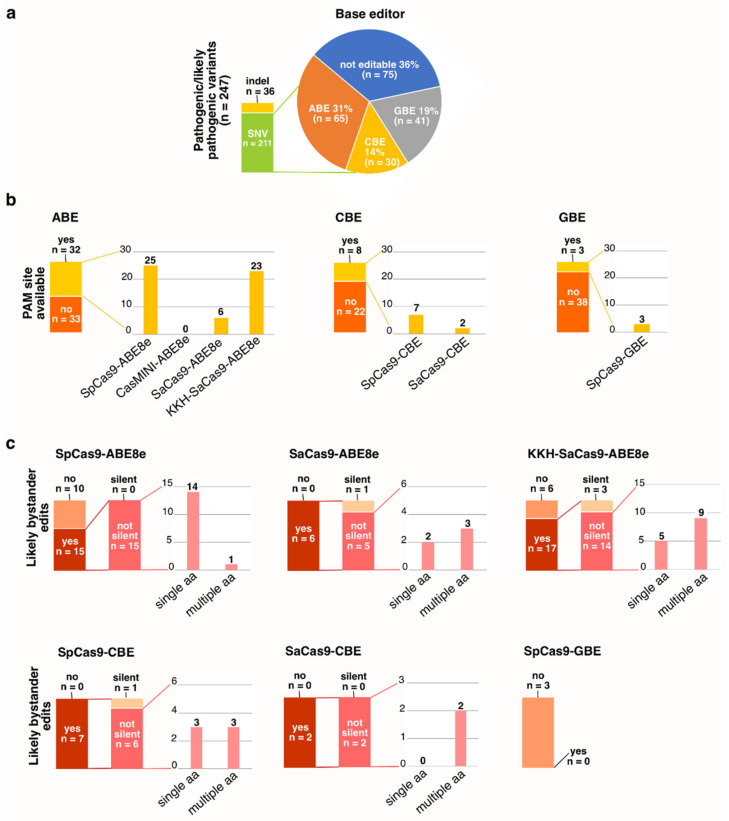
Base editing options for disease-associated *RHO* variants. (**a**) LOVD, ClinVar and gnomAD datasets included altogether 247 unique *RHO* variants that were interpreted as pathogenic or likely pathogenic in at least one of the datasets. Judging by the variant type, 31% of the SNVs could be corrected with ABE, 19% with GBE and 14% with CBE. (**b**) Available PAM sites were found for 49% of the ABE-editable variants, 27% of the CBE- and 7.3% of the GBE-editable variants. Available PAM sites were more commonly found for SpCas9-ABE8e and KKH-SaCas9-ABE8e constructs, which enabled guide RNA design for 25 and 23 disease-associated variants, respectively. (**c**) Bystander edits were likely to be introduced in the majority of guide RNAs for all constructs except the SpCas9-GBE. Amongst predicted bystander edits, amino acid-changing bystanders were more common than silent variants for all constructs. The majority of SpCas9-ABE8e-guides were likely to introduce bystander edits that result in one amino acid being changed, whereas bystander edits in SaCas9-ABE8e-, KKH-SaCas9-ABE8e- and SaCas9-CBE-guides were predicted to alter multiple amino acids.

## Data Availability

All data analyzed in this study are publicly available through the LOVD, ClinVar and gnomAD databases, and all resulting analyzed data are included in this manuscript as Appendix A.

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
