# Peer review of "CRISPR DNA Base Editing Strategies for Treating Retinitis Pigmentosa Caused by Mutations in Rhodopsin"

_genes, 2022, doi:10.3390/genes13081327_

Round 1

Reviewer 1 Report

In the present manuscript Kaukonen et al. provide a systematic analysis of reported pathogenic RHO variants in three commonly used human genetic variation databases, dissecting out the potential of current base editors to correct them.

While the proposed study is in line with current efforts to find curative solutions to prevalent inherited diseases through the use of state-of-the-art gene editing technologies, it is of questionable significance and does not provide any advancements of the current knowledge in the field. The data presented in this study could be a good complement to a literature review, such as the one referenced by the authors (ref 20), but feels insufficient to be published as a stand-alone scientific manuscript.

Major concerns:

·         >I do not see what the benefit is of repeating the same analysis separately for the three databases, apart from just extending the length and redundancy of the study.  Figures 2-4 could be best represented as a combined analysis if the variants in the three databases, which is what you do on results section 3.4.

·          >Presence of a PAM site is indispensable for base editing activity. Therefore, considering the editability of certain genetic variants just based on the target sequences misleading and incorrect. Figures 2c, 3c and 4c are not correctly representing the editability of the target site as they do not take presence of the PAM site into account.

·         >Base editors are a very interesting technology that hold big promises for correction of SNV-associated pathologies. However, in order to provide the article with more relevance, authors should consider the editability of the variants also with newer and more versatile gene editing tools, such as Prime Editing.

·         >Material and methods sections is too concise. Authors should include more details on the bioinformatic tools used to design sgRNA, predict off-target sites, bystander edits, etc.

·         >The manuscript would be more interesting to the scientific community if some proof-of-concept experimental results were performed in line with the descriptive data presented. E.g. base editing iPSC from a patient with one the most common genetic variants.

Minor concerns:

·         >Line 54: When mentioning that Cas protein on BE is deactivated, authors should take into account that some BE use a nickase form of Cas but not a catalytically dead one.

·         >When designing sgRNAs, it would be good to compare at least 2 other bioinformatic tools other than Benchling

·        >Line 138: Why c.68C>A, p.Pro23His variant cannot be corrected? Is not that mutation amenable for GBE base editing?

·        >Color selection used in pie charts of the different figures should be harmonized. E.g. orange color on Fig 2b, 3b and 4b should always correspond to VUS.

·         >Why there is no figure representing the most common variants in ClinVar and gnomAD (similar to figure 2d, referring to LOVD)?

·         >Line 198: Should specify: “not editable with BE

Author Response

In the present manuscript Kaukonen et al. provide a systematic analysis of reported pathogenic RHO variants in three commonly used human genetic variation databases, dissecting out the potential of current base editors to correct them. While the proposed study is in line with current efforts to find curative solutions to prevalent inherited diseases through the use of state-of-the-art gene editing technologies, it is of questionable significance and does not provide any advancements of the current knowledge in the field. The data presented in this study could be a good complement to a literature review, such as the one referenced by the authors (ref 20), but feels insufficient to be published as a stand-alone scientific manuscript.

Response: We thank the reviewer for taking the time to read our manuscript, however we disagree with the argued lack of significance of our study. Rhodopsin mutations are the most common cause for autosomal dominant retinitis pigmentosa, a clinically important blinding eye disease. However, as we describe in the introduction section, safe and effective treatment is still lacking. DNA base editing has many advantages (lines 48-67) that makes it promising technology to treat dominant mutations. As we show in the article, with extensive and thorough in silico analyses, base editing could correct over half of the disease associated RHO-variants. Importantly, we also discuss the technology’s current weaknesses (discussion section, concluded on lines 349-352) to highlight the important areas in need of further improvements before the technology can be widely utilized to treat RHO-associated adRP or other diseases.

Major concerns:

  1. I do not see what the benefit is of repeating the same analysis separately for the three databases, apart from just extending the length and redundancy of the study. Figures 2-4 could be best represented as a combined analysis if the variants in the three databases, which is what you do on results section 3.4.

Response: We thank the reviewer for this comment, however, as we describe on lines 254-272, the data from the utilized three databases have important differences in terms of, for example, set-up (patient data in LOVD and ClinVar versus population data in gnomAD; different accompanying information) and content (some disease-associated variants are reported in only one of the three datasets, lines 265-268), and hence we conclude that including data from all of the three datasets will maximize the number of important variants covered in this study. As is reported in the dataset-specific results sections (lines 125-190) and discussion (273-285), interesting comparisons between the three datasets can also be done regarding the variant prevalence, which is important aspect when choosing target variants for future base editing experiments.

  1. Presence of a PAM site is indispensable for base editing activity. Therefore, considering the editability of certain genetic variants just based on the target sequences misleading and incorrect. Figures 2c, 3c and 4c are not correctly representing the editability of the target site as they do not take presence of the PAM site into account.

Response: We thank the reviewer for this comment, however, as new base editing constructs utilising new nucleases with varying PAM site requirements are developed with increasing pace, it is likely that majority of the variants that are of editable type, but currently without available PAM sites, will become targetable in the near future. Such constructs could be e.g. CasPhi with very flexible TBN-PAM site (discussed on line 352), or others, as multiple leading groups are working hard to develop new constructs. Therefore, we feel it is important to note that indeed majority of disease-associated variants could the edited with future constructs, but further development of the constructs is needed – as discussed in the manuscript.

  1. Base editors are a very interesting technology that hold big promises for correction of SNV-associated pathologies. However, in order to provide the article with more relevance, authors should consider the editability of the variants also with newer and more versatile gene editing tools, such as Prime Editing.

Response: We thank the reviewer for this comment and have included discussion on prime editing and CRISPR/Cas9 “knock-out-and-replace” approach as examples of other potentially useful methods to correct disease-associated dominant point mutations and indels (lines 355-366). However, these methods have also important limitations: as an example, prime editing in its current form has significantly reduced editing efficiencies compared to base editing and none of the current constructs fit into a single AAV, the preferred delivery method for photoreceptors. For these reasons, and to ensure clarity and appropriate length of the manuscript, we decided to focus on DNA base editing in this study.

  1. Material and methods sections is too concise. Authors should include more details on the bioinformatic tools used to design sgRNA, predict off-target sites, bystander edits, etc.

Response: We thank the reviewer for this comment. As described in the methods section, the sgRNAs were designed using the Benchling software with the construct-specific parameters illustrated in Figure 1 and utilising the specified RHO reference sequence. All guide designing softwares are in principle text mining tools searching the DNA sequence for desired nucleotide motifs for specified PAM sites, and hence all guide design softwares produce identical results unless clearly faulty in design (which Benchling is not). Once the software has listed available PAM sites, bystander edits can easily be analysed manually utilising codon tables. Predicting off-target edits might be applicable method to prioritise multiple sgRNAs, and we agree that for that specific purpose, comparing multiple in silico softwares would be informative. However, for this study, it is not useful, as in majority of cases only one sgRNA was available (Table S4) and hence no prioritising could be done.

  1. The manuscript would be more interesting to the scientific community if some proof-of-concept experimental results were performed in line with the descriptive data presented. E.g. base editing iPSC from a patient with one the most common genetic variants.

Response: We thank the reviewer for this comment and agree that including proof-of-concept data would be interesting. However, the aim of this study was to analyse the current potential and limitations in broader gene-level aspect and not limit analysis for a specific variant, as well as to provide a primary review of the future of base editing to treat RHO-associated adRP. Future work in the coming years will show variant-specific applications.

Minor concerns:

Line 54: When mentioning that Cas protein on BE is deactivated, authors should take into account that some BE use a nickase form of Cas but not a catalytically dead one.

Response: We thank the reviewer for pointing this out and have revised the manuscript accordingly.

When designing sgRNAs, it would be good to compare at least 2 other bioinformatic tools other than Benchling

Response: We thank the reviewer for this comment, however, as discussed above (point 4.), different guide design softwares do not produce different results in terms of guide sequence design and prioritising multiple guides based on the predicted off-target effects was not applicable (point 4.).

Line 138: Why c.68C>A, p.Pro23His variant cannot be corrected? Is not that mutation amenable for GBE base editing?

Response: We thank the reviewer for this comment. Unfortunately, as a C-to-A transversion, the current base editors (ABEs, CBEs, GBEs) are unable to correct this mutation. The GBEs can introduce G-to-C and C-to-G edit only (line 90).

Color selection used in pie charts of the different figures should be harmonized. E.g. orange color on Fig 2b, 3b and 4b should always correspond to VUS.

Response: We thank the reviewer for this very helpful comment and have harmonized the figures as suggested to improve clarity.

Why there is no figure representing the most common variants in ClinVar and gnomAD (similar to figure 2d, referring to LOVD)?

Response: We thank the reviewer for this comment. In the ClinVar dataset, allele frequency information is not provided: each variant is provided only with the information whether it has been reported once or multiple times (Table S2). For the gnomAD dataset, allele frequencies are provided for each variant, but as they differ significantly between populations (Table S3), we felt it was more informative to describe the details in result text (lines 179-183) and discuss them in relation to the LOVD data (lines 273-285), whereas creating a figure might easily be misleading. We hope the readers will find this approach helpful.

Line 198: Should specify: “not editable with BE”

Response: We thank the reviewer for this helpful comment and have revised the manuscript accordingly.

Reviewer 2 Report

CRISPR DNA base editing strategies for treating retinitis pigmentosa caused by mutations in Rhodopsin

Kaukonen et al.,

In this study, the authors conducted an analysis of several base editors and the extent to which these editors can be used to correct Rhodopsin (RHO) mutations that cause autosomal dominant retinitis pigmentosa (adRP). I enjoy reading the details. I only have some minor suggestions for the authors to consider.

1.       Some of the PAM discussions in the Results and Discussion sessions are repetitive and can potentially be combined and streamlined.

2.       I understand the genome-editing field is evolving quickly and no paper should include all the latest details. However, since this manuscript is discussing new genome-editing treatments for RHO-associated adRP, I thought a brief discussion and comparison between different proposed approaches would be useful for the audience to put the base-editing approaches in context. What approach would the authors use if they were asked to choose an approach now for the next clinical trial? For example,

a.       The ablation and replacement approach as proposed by Editas < https://www.editasmedicine.com/wp-content/uploads/2020/05/Diner_Dass_ASGCT_2020-RhoADRP_Final.pdf>. I understand the authors described on lines 45-46 that this generic approach is “limited” because of the requirement for silencing the defective allele. However, this approach can also be exciting as it can be a standardized, mutation-independent therapy and will be cheaper for patients.

b.       The prime-editing approach, which may confer some advantages and flexibilities over the base-editing approaches. I understand the author wrote a review that mentioned this approach (Kantor et al., 2020) and am certain that they are aware of this approach.

c.       The near-PAMless Cas9 base editors and how they might have partially resolved the PAM issues that the author raised (Walton et al., Science. 2020;368:290; and other follow-up studies).

3.       Typo: Line 127: 7.9%?

Author Response

In this study, the authors conducted an analysis of several base editors and the extent to which these editors can be used to correct Rhodopsin (RHO) mutations that cause autosomal dominant retinitis pigmentosa (adRP). I enjoy reading the details. I only have some minor suggestions for the authors to consider.

Response: We thank the reviewer for taking the time to read our manuscript and the very positive comments provided.

  1. Some of the PAM discussions in the Results and Discussion sessions are repetitive and can potentially be combined and streamlined.

Response: We thank the reviewer for this helpful comment and have revised the manuscript accordingly (lines 301-304).

  1. I understand the genome-editing field is evolving quickly and no paper should include all the latest details. However, since this manuscript is discussing new genome-editing treatments for RHO-associated adRP, I thought a brief discussion and comparison between different proposed approaches would be useful for the audience to put the base-editing approaches in context. What approach would the authors use if they were asked to choose an approach now for the next clinical trial? For example,
  2. The ablation and replacement approach as proposed by Editas < https://www.editasmedicine.com/wp-content/uploads/2020/05/Diner_Dass_ASGCT_2020-RhoADRP_Final.pdf>. I understand the authors described on lines 45-46 that this generic approach is “limited” because of the requirement for silencing the defective allele. However, this approach can also be exciting as it can be a standardized, mutation-independent therapy and will be cheaper for patients.
  3. The prime-editing approach, which may confer some advantages and flexibilities over the base-editing approaches. I understand the author wrote a review that mentioned this approach (Kantor et al., 2020) and am certain that they are aware of this approach.
  4. The near-PAMless Cas9 base editors and how they might have partially resolved the PAM issues that the author raised (Walton et al., Science. 2020;368:290; and other follow-up studies).

Response: We thank the reviewer for this excellent suggestion and have revised the manuscript accordingly (lines 310-314 and 355-366).

  1. Typo: Line 127: 7.9%?

Response: We thank the reviewer for pointing this out and have revised the manuscript accordingly.

Reviewer 3 Report

This is a well written and interesting manuscript reporting the strategies and effectiveness of base editing for the correction of Rhodopsin associated pathogenic variants. Authors have taken extensive efforts to evaluate the potential of currently available base editors, guide sequences for their PAM availability and likely bystander edits for 55% of pathogenic and likely pathogenic rhodopsin variants.  The study provides useful information for future studies that can further clarify the basis for both in vitro and in vivo studies. Additionally, the figures and the supplementary data fully support the conclusions reached except for a minor exception below.

Minor points:

Line number 127:  The majority (92%) of the reported variants were SNVs, while 7,9% were indels. The percentage of indels is mistakenly indicated. 

Author Response

This is a well written and interesting manuscript reporting the strategies and effectiveness of base editing for the correction of Rhodopsin-associated pathogenic variants. Authors have taken extensive efforts to evaluate the potential of currently available base editors, guide sequences for their PAM availability and likely bystander edits for 55% of pathogenic and likely pathogenic rhodopsin variants. The study provides useful information for future studies that can further clarify the basis for both in vitro and in vivo studies. Additionally, the figures and the supplementary data fully support the conclusions reached except for a minor exception below.

Response: We thank the reviewer for taking the time to read our manuscript and the very positive comments provided.

Minor points:

Line number 127:  The majority (92%) of the reported variants were SNVs, while 7,9% were indels. The percentage of indels is mistakenly indicated.

Response: We thank the reviewer for pointing this out and have revised the manuscript accordingly.

Round 2

Reviewer 1 Report

I thank the authors for considering some of my comments and incorporating some of the modifications I suggested. However, I still consider that the present manuscript lacks experimental weight to be published in the form a scientific article and not as a review or commentary.

The analysis of frequent variants described in databases is part of the job that all (or most) of the investigators that work on heritable diseases perform prior to initiate our scientific proyects related to a specific disease, whether is genome editing, disease modelling or molecular/cellular profilling. The "lack of significance" that I mentioned on my first comment does not refer to the disease you chose to investigate, which I personally consider of major importance, but instead to the significance of the content included in the manuscript.

Nevertheless, if the other two reviewers and the editor consider that the present manuscript is adequated for publication I will not oppose to that.